# Ethno-Veterinary Survey and Quantitative Study of Medicinal Plants with Anthelmintic Potential Used by Sheep and Goat Breeders in the Cotton Zone of Central Benin (West Africa)

Christian Cocou Dansou [1], Pascal Abiodoun Olounladé [1,*], Basile Saka Boni Konmy [1], Oriane Songbé [1], Kisito Babatoundé Arigbo [1], André Boha Aboh [1], Latifou Lagnika [2] and Sylvie Mawulé Hounzangbé-Adoté [3]

[1] Zootechnical Research and Livestock System Unit, Laboratory of Animal and Fisheries Science (LaSAH), National University of Agriculture (UNA), Porto-Novo 01 BP 55, Benin; chrisdansou@yahoo.fr (C.C.D.); bsakabasile.konmy@uac.bj (B.S.B.K.); sooria@etu.una.bj (O.S.); arigbokisito@gmail.com (K.B.A.); aboh.solex@gmail.com (A.B.A.)
[2] Laboratory of Biochemistry and Bioactive Natural Substances, Faculty of Science and Technology, University of Abomey-Calavi, Cotonou 01 BP 526, Benin; latifkabe@yahoo.fr
[3] Laboratory of Ethnopharmacology and Animal Health, Faculty of Agronomic Sciences, University of Abomey-Calavi, Cotonou 01 BP 526, Benin; syladote@yahoo.fr
* Correspondence: polounlade@gouv.bj; Tel.: +229-97-08-54-68

**Abstract:** This study presents the diversity of anthelmintic plants in the cotton zone of Central Benin. The aim was to identify the medicinal anthelmintic plants used by small ruminant breeders in cotton zone of Central Benin to treat gastrointestinal parasites. Three hundred and sixty breeders were selected during individual semi-structured face-to-face interviews. Different quantitative indices of cultural importance were calculated in order to determine the level of use of plant species. Jaccard similarity index (JI) was calculated and Pearson's correlation was determined for Use Value (UV) and Relative Frequency of Citation (RFC). In this study, a total of 99 medicinal species, of which 63 have anthelmintic potential, were listed, including *Khaya senegalensis*, *Launaea taraxacifolia*, *Napoleonaea vogelii*, *Momordica charantia* and *Vernonia amygdalina*, which all had UV and RFC above 20%. Each of them had a Fidelity Level above 50% and an Informant Agreement Rate (IAR) value close to one. Pearson's correlation showed a significant correlation between RFC and UV with r = 0.94, and the studies were clearly independent (IJ < 50%). This study showed that the cotton zone of Central Benin represents 4% of the total flora of Benin, with many anthelmintic plants such as *Launaea taraxacifolia* and *Napoleonaea vogelii* that require further investigation.

**Keywords:** anthelmintic plants; ethno-veterinary; floristic richness; gastrointestinal parasites; Benin

## 1. Introduction

The development of small ruminant breeding and the improvement of their zootechnical characteristics constitute a major global challenge. Benin, one of the West African countries where small ruminant farming is one of the main activities of rural populations, to satisfy their needs for meat products and money, shares these concerns. With a surface area of 114.763 km$^2$, it is a country located on the coastal strip of the Golf of Guinea between the parallels 6°10′ and 12°25′ of northern latitudes and the meridians 0°45′ and 3°55′ of eastern longitudes, and is bounded to the north by Niger, to the south by the Atlantic Ocean, to the east by Nigeria and to the west by Togo and Burkina Faso. On the basis of climate, relief, soils, vegetation cover, livestock type and land form, Benin is defined in several zones [1], including the cotton zone of Central Benin, known for its large livestock population. It alone accounts for more than half of the national livestock population [2]. For the population, livestock farming is a profitable activity that allows them to ensure food security [3,4], but the sector remains underdeveloped and confronted with numerous pathologies, including parasitic diseases [5]. To cope with this situation, breeders are

turning to traditional veterinary medicine given the increasingly proven ineffectiveness of synthetic anthelmintics, the problems of resistance and the consequences of their use on human health and the environment. It is this situation that motivated us to identify the medicinal plants used by breeders in the cotton zone of Central Benin to treat gastrointestinal parasites in small ruminants. Several ethno-veterinary studies have been conducted in Benin [6–9], but the studies carried out remain partial because the cotton zone of Central Benin is not fully taken into account in the documentation of plants with therapeutic and anthelmintic potential in particular. The objective of the present study is to identify the medicinal anthelmintic plants used by small ruminant breeders in the cotton zone of Central Benin to treat gastrointestinal parasites of small ruminants.

## 2. Materials and Methods

### 2.1. Description of the Study Area

The study was carried out in the cotton zone of Central Benin. This zone is the largest and includes the entire Collines department and parts of the Borgou, Donga, Couffo, Plateau and Zou departments. The ethnobotanical survey was carried out in the Communes of Bassila, Parakou, Tchaourou, Ouesse, Bante, Save, Glazoue, Ketou, Djidja, Dassa-Zoume, Savalou and Aplahoue (Figure 1). The zone covers an area of 32.163 km$^2$ and is characterised by a Sudano-Guinean climate. It is an area characterised by sheep, goat, cattle, pig and poultry farming [10].

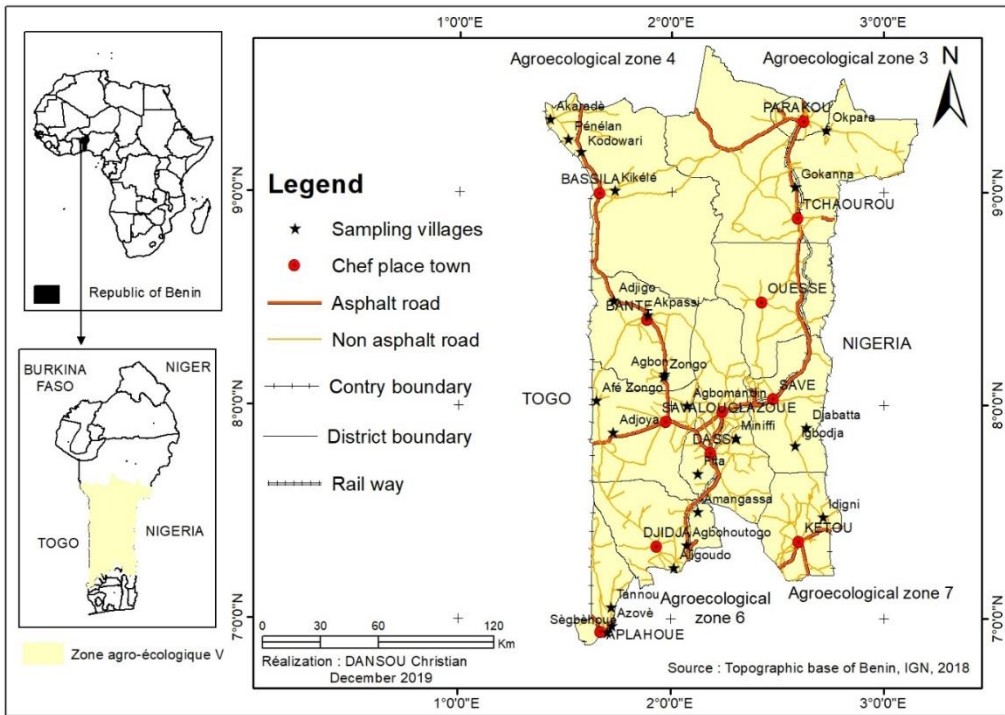

**Figure 1.** Map of the cotton zone of Central Benin.

### 2.2. Exploratory Study

The selection of respondents was random and simple. To this end, an exploratory study was conducted beforehand over a period of three weeks. This survey made it possible to validate our questionnaire and to determine the proportion (p) of small ruminant breeders who responded that they were aware of or used medicinal plants to prevent or treat parasitic diseases in small ruminants in their livestock. The number of respondents was determined using the formula of Dagnélie [11]:

$$n = \frac{P(1-P)U^2_{1-\alpha/2}}{d^2}$$

where $n$ is the overall sample size; $p$ is the proportion of breeders in the survey area who use or are aware of medicinal plants used in the prevention or treatment of small ruminant gastrointestinal parasites; $U_{1-\alpha/2}$ is the value of the random normal variable for a probability value of $\alpha$; $U_{1-\alpha/2} = 1.96$ if $\alpha = 0.05$; and $d$ is the marginal error of 5%.

### 2.2.1. Inclusion Criteria

Small ruminant breeders living in the different communes of the study area and that use or know the medicinal plants to treat pathologies; in particular, gastrointestinal parasite of small ruminants.

### 2.2.2. Exclusion Criteria

The following were excluded from our study: breeders visiting the commune and not residing in the study area; breeders without knowledge of the use of medicinal plants to treat pathologies in sheep and goats; those who had not given their consent to participate in the survey and those who had been visited without success. Additionally, excluded from the study were those who were aphonic or unable to communicate.

### 2.3. Data Collection

Data were collected between November 2019 and January 2020 following semi-structured individual interviews and direct observations with the assistance of a local translator where necessary. The basic questionnaire was structured in two parts. The first part covered socio-demographic characteristics (age, gender and main activity). The second part concerned medicinal plants in general and those with anthelmintic potential, their preparation methods, their administration route and the parts used were indicated. In the data collection, only herbal treatments were considered in this study. Remedies involving the use of ingredients such as potash, salt or others were not considered. The medicinal plants mentioned were identified in the area where breeders usually collect them with note for each plant species collected. Specimens were collected and numbered on the spot, later identified using Benin's analytical flora, according to Akoègninou et al. [12]. Additional identification was carried out by matching specimens with those previously identified held in the National Herbarium of Benin. Photographic and video cameras were used for graphic documentation. The botanical names of the plant specimens were updated according to The World Flora Online. The first breeders who met the inclusion criteria were identified and randomly selected in each commune of the study area with the help of the Heads of Communal Cells of the Territorial Agency for Agricultural Development. Then, the non-probabilistic 'snowball' method [13] was used to identify the following breeders. In this method, the first breeders surveyed indicated other small ruminant breeders and thus became additional informants. All those identified in an area who met the inclusion criteria were surveyed. The ethnoveterinary survey was conducted among 360 breeders in the study area, i.e., 30 respondents per commune.

### 2.4. Data Processing and Statistical Analyses

The collected ethnobotanical data were entered and organised using both Microsoft Excel 2016 and the statistical software Sphinx Plus² (V5), in order to analyse and identify the frequencies and percentages of the respondents' socio-demographic data and the number of plants used to treat certain ailments in small ruminants, as well as the different proportions of plant parts used and the botanical families to which they belonged. The results of the ethnobotanical survey were analysed using various quantitative indices of relative cultural importance to determine the level of knowledge and use of plant species by breeders/agro-pastoralists. These included Relative Frequency of Citation (RFC), Informant Consensus Factor (ICF), Informant Agreement Rate (IAR) and Fidelity Level (FL). These different indices were calculated in order to assess the importance of the recorded plant species and to understand the potential use of each species. The different indices calculated were based on the principle that plants that had a high citation were considered to be the most

significant, and therefore were much more important than those that had low citations. The different ethnobotanical indices were calculated using the following statistical formula:

- Relative Frequency of Citation

The Relative Frequency of Citation (RFC) expressed as a percentage refers to the number of respondents (n) using or having knowledge of a given medicinal plant, compared to the total number of respondents (N). It was used to estimate the local importance of the species cited [14,15].

$$RFC = \frac{n}{N} \times 100$$

- Use Value

The Use Value (UV) of species for the medicinal plants is calculated according to the formula [16]:

$$UV = \frac{U}{N} \times 100$$

where U is the number of times a species is cited and N is the total number of informants interviewed. This index was used to measure the relative degree of use of each of the plants cited by the breeders/agro-pastoralists. In contrast with the rarity index of Géhu and Géhu [17], plants with a usual value of more than 20% were considered to be preferred plants, and therefore widely used for the treatment of gastrointestinal parasites. Otherwise, they were less used.

- Fidelity Level (FL)

The percentage of informants who reported using a certain plant species for the same main purpose was calculated for the most frequently reported diseases or conditions. This index from Friedman et al. [18] is calculated as follows:

$$FL = \frac{n}{nt} \times 100$$

With n = frequency of citation of the species in the treatment of a particular condition and nt = total number of citations of the species.

- Informant Agreement Rate (IAR)

This index allows the identification of species with a significant therapeutic index among the respondents. It was calculated according to the formula [19]:

$$IAR = \frac{(Nr - Na)}{(Nr - 1)}$$

where Nr = total number of citations of the species and Na = number of diseases treated by the species.

- The Informant Consensus Factor (ICF)

This measured the degree of homogeneity of knowledge among informants for the diseases. It was calculated according to the formula [20]:

$$ICF = \frac{(Nur - Nt)}{(Nur - 1)}$$

where Nur is the total number of citations and Nt is the number of species cited. IFC is between 0 and 1. The more ICF tends towards 1, the more consensus there is among informants.

- Jaccard index (JI)

We also wished to calculate similarities between our studies with other studies carried out in other parts in Benin and neighbouring countries. This may be expressed using the Jaccard similarity index (JI), which uses the following formula [21,22]:

$$JI = 100 \times \frac{C}{(a + b - c)}$$

where c is the number of species common to both areas; a is the number of species exclusive in another studies and b is the total number of species in the present study. If IJ > 50%, the studies are similar and if IJ < 50%, there is dissimilarity between the studies. In practice, when IJ > 45%, it is accepted that there is similarity between the study areas concerned.

Multivariate analyses were also performed using R for data science version 3.6.3, USA. The multivariate analysis consisted of a Principal Component Analysis (PCA), performed on the relative frequency of citation plants for different diseases to examine whether the citing was consistent across the survey areas. This provided a representation of the diseases and species cited as projections onto planes defined by the first factorial axes. A Hierarchical Ascending Classification (HAC), according to Ward's method, using the Euclidean distance, made it possible to group the farms according to the traditional practices used by the subjects. All the plants used were represented in the form of a dendrogram.

## 3. Results

### 3.1. Socio-Demographic Characteristics of Respondents

Our study involved 360 breeders in all the 12 communes of the study area. Each of the respondents was questioned by face-to-face interviews to collect data on their breeding practices and the use of medicinal plants to treat diseases, in particular gastrointestinal parasitosis in small ruminants. The study population was predominantly male (70.8%), with females representing 29.2%. The age of the respondents ranged from 20 to 80 years with an average age of $45.85 \pm 12.68$. The majority of respondents were farmers (58.88%), followed by breeders (21.66%) and traders (10.58%). In most cases, respondents have dual (farmer–breeder) or multiple activities, with the two farmer–breeder activities often linked. The other activities (artisans, traditherapists, civil servant, drivers, resellers and veterinarians) represented less than 5% (Table 1).

**Table 1.** Characteristics of respondents.

| Characteristics | Number (*n* = 360) | Percentage (%) |
|---|---|---|
| **Gender** | | |
| Male | 255 | 70.8 |
| Female | 105 | 29.2 |
| **Age group** | | |
| [20–40[ | 118 | 32.8 |
| [40–60[ | 185 | 51.4 |
| ≥60 | 57 | 15.8 |
| **Main activity** | | |
| Farmers | 212 | 58.89 |
| Breeders | 78 | 21.66 |
| Traders | 38 | 10.53 |
| Artisans | 12 | 3.31 |
| Resellers | 6 | 1.71 |
| Civil servant | 5 | 1.40 |
| Traditherapists | 4 | 1.10 |
| Drivers | 4 | 1.10 |
| Veterinarians | 1 | 0.30 |

### 3.2. Taxonomic Diversity of Medicinal Plants with Therapeutic Values

At the end of the survey, a total of 99 plant species were counted as medicinal plants used by livestock breeders in the study area to treat the health problems of small ruminants

(Table 2), of which 63 were used as anthelmintic plants. These species identified represented 88 genera and 43 botanical families. The Leguminosae were the most represented family with twelve species. Next came the Euphorbiaceae with ten species, the Poaceae with six species, then the Anacardiaceae, Cucurbitaceae, Malvaceae, Moraceae, Rubiaceae and Rutaceae were each represented by four plant species. The other families were less represented (Figure 2). The majority of genera (80) were represented by a single species. Six genera were represented by two species. These were *Annona*, *Caesalpinia*, *Manihot*, *Senna*, *Ocimum* and *Piliostigma*. The most represented genera were *Ficus* with four species, followed by *Citrus* (three species).

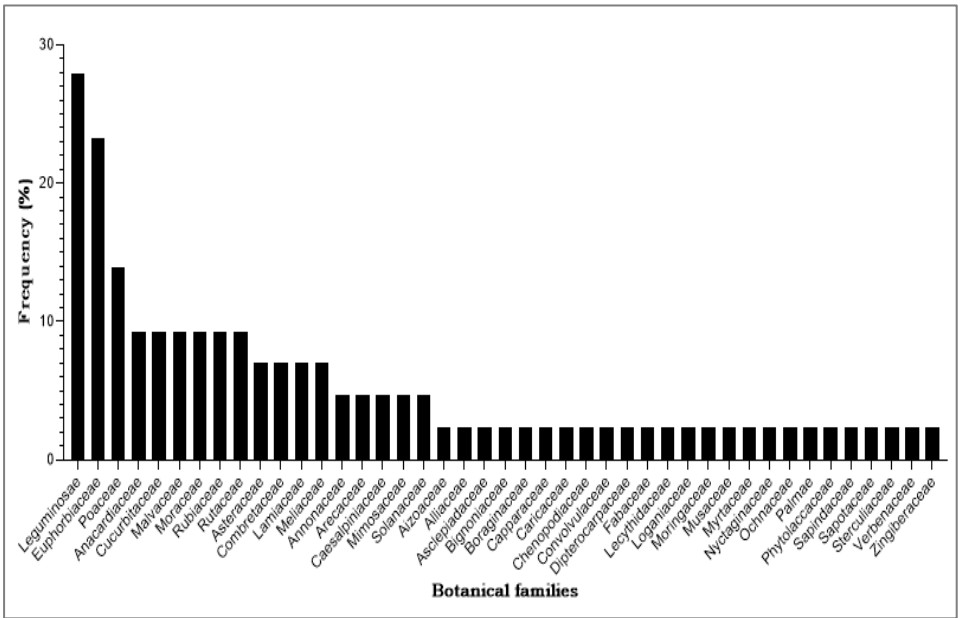

**Figure 2.** Distribution of medicinal plants by botanical family.

*3.3. Plant Parts Used, Method of Preparation and Route Administration*

The breeders surveyed used different parts of the plants for the preparation of remedies. Leaves were the most commonly used (55.99%) in the process of livestock care, followed by bark (21.60%), leafy stem (5.60%), whole plant (4%), fruit (4%), root (1.60%), seed (1.60%), spike (0.8%) and bulb (0.80%) (Table 3). In the healing process, the preparation of the remedies was extemporaneous and usually monospecific. Regarding the form in which the remedies were used, the feeding mode in the form of fodder (fresh leaves) was in first position with 49.01% (Table 4), followed by decoction with 20.53%. Remedy intake in the form of powder incorporated in the feed or drinking water and triturate form accounted for 13.25% and 7.28%, respectively. The other methods of preparation were maceration (5.30%), infusion (3.97%) and roasted form (0.66%). These plant species with therapeutic value used for the treatment of small ruminant diseases in the form of leaves for grazing or in other traditional forms were mainly administered orally, with a percentage of 94.18% (Table 4). The other routes of administration were instillation (2.91%), cutaneous (1.94%) and nasal route (0.97%).

**Table 2.** Diversity of medicinal plants used by breeders in the cotton zone of Central Benin.

| N° | Species | Families | Parts Used | Preparation Methods | Administration Route | Diseases Treated | UV |
|----|---------|----------|------------|---------------------|----------------------|------------------|-----|
| 1 | *Vernonia amygdalina* Delile | Asteraceae | L | Tri | Oral | IP, DD, EP, BD, RD, Sym | 0.486 |
| 2 | *Khaya senegalensis* (Desr.) A.Juss. | Meliaceae | L,B | Mac, Inf, Dec, Fod, Powder | Oral | IP, DD, EP, BD, Sym | 0.436 |
| 3 | *Launaea taraxacifolia* (Willd.) Amin ex C. Jeffrey | Asteraceae | L, Ls | Mac, Inf, Dec, Fod, Powder | Oral | IP, DD, RD | 0.431 |
| 4 | *Napoleonaea vogelii* Hook. & Planch | Lecythidaceae | L | Mac, Inf, Dec, Fod, Powder | Oral | IP, DD | 0.378 |
| 5 | *Momordica charantia* L. | Cucurbitaceae | L | Mac, Powder | Oral | IP, DD, EP, BD | 0.358 |
| 6 | *Moringa oleifera* Lam. | Moringaceae | L, Ls | Fod | Oral | IP, DD, EP, RD, Sym | 0.356 |
| 7 | *Zanthoxylum zanthoxyloides* (Lam.) Zepern. & Timler | Rutaceae | L | Fod | Oral | IP, DD, EP, RD | 0.269 |
| 8 | *Adansonia digitata* L. | Malvaceae | L,B | Mac, Inf, Dec, Fod, Powder | Oral | DD, BD, RD | 0.197 |
| 9 | *Morinda lucida* Benth. | Rubiaceae | L | Fod | Oral | IP, DD, BD, RD, Sym | 0.197 |
| 10 | *Detarium microcarpum* Guill. & Perr. | Caesalpiniaceae | B | Dec | Oral | IP | 0.172 |
| 11 | *Ocimum gratissimum* L. | Lamiaceae | L | Fod | Oral | IP, EP, BD, Sym | 0.164 |
| 12 | *Azadirachta indica* A. Juss. | Meliaceae | L,B,S | Mac, Dec, Fod, Powder | Oral | IP, DD, EP, BD, Sym | 0.147 |
| 13 | *Carica papaya* L. | Caricaceae | L,S | Fod, Powder | Oral | IP, EP | 0.144 |
| 14 | *Spondias mombin* L. | Anacardiaceae | L | Fod | Oral | DD, EP, RD | 0.128 |
| 15 | *Pterocarpus erinaceus* Poir. | Leguminosae | L,B | Fod, Powder | Oral | IP, BD, RD | 0.125 |
| 16 | *Elaeis guineensis* Jacq. | Arecaceae | L | Fod | Oral | IP, DD, EP | 0.111 |
| 17 | *Lannea acida* A. Rich | Anacardiaceae | B | Dec | Oral | DD | 0.108 |
| 18 | *Newbouldia laevis* Seem. ex Bureau | Bignoniaceae | L | Fod | Oral | IP, DD, BD, RD | 0.108 |
| 19 | *Leucaena leucocephala* (Lam.) de Wit | Mimosaceae | L | Fod | Oral | DD, EP, RD, Ap, Sym | 0.092 |
| 20 | *Annona senegalensis* Pers. | Annonaceae | L,B,R | Dec, Fod, Powder | Oral | IP, DD, PE, RD | 0.083 |
| 21 | *Caesalpinia bonduc* (L.) Roxb. | Leguminosae | L | Fod | Oral | IP, BD, RD | 0.064 |
| 22 | *Sporobolus pyramidalis* P.Beauv. | Poaceae | Ls, Wp | Fod | Oral | Ap | 0.058 |
| 23 | *Hyptis suaveolens* (L.) Poit. | Lamiaceae | L | Tri | Oral | IP, DD | 0.056 |
| 24 | *Piliostigma reticulatum* (DC.) Hochst. | Leguminosae | L | Fod | Oral | IP | 0.056 |
| 25 | *Senna alata* (L.) Roxb. | Leguminosae | L | Fod, Powder | Oral | IP, EP, Sym | 0.047 |
| 26 | *Sterculia setigera* Delile | Sterculiaceae | L,B | Fod, Powder | Oral | IP | 0.047 |
| 27 | *Chromolaena odorata* (L.) R.King & H.Rob. | Asteraceae | L | Fod | Oral | IP, DD, EP, BD, RD, Sym | 0.044 |
| 28 | *Piliostigma thonningii* (Schumach.) Milne-Redh. | Leguminosae | L | Dec, Fod | Oral | IP, EP, RD, Sym | 0.044 |
| 29 | *Terminalia avicennioides* Guill. & Perr. | Combretaceae | R | Dec, Tri | Oral, Instillation | DD, EP, K | 0.044 |
| 30 | *Petiveria alliacea* L. | Phytolaccaceae | L | Fod | Oral | Ap | 0.039 |
| 31 | *Phyllanthus amarus* Schumach. & Thonn. | Euphorbiaceae | Ls | Fod | Oral | DD | 0.039 |
| 32 | *Zea mays* L. | Poaceae | Epi | Roast | Dermal route | IP, DD | 0.039 |
| 33 | *Ficus exasperata* Vahl | Moraceae | L | Fod | Oral | DD, RD | 0.036 |
| 34 | *Sarcocephalus latifolius* (Sm.) E.A.Bruce | Rubiaceae | B | Dec | Oral | IP, EP | 0.036 |

Table 2. *Cont*.

| N° | Species | Families | Parts Used | Preparation Methods | Administration Route | Diseases Treated | UV |
|---|---|---|---|---|---|---|---|
| 35 | *Mallotus oppositifolius* (Geiseler) Müll.Arg. | Euphorbiaceae | L, Ls | Fod | Oral | DD, EP | 0.033 |
| 36 | *Mitragyna inermis* (Willd.) Kuntze | Rubiaceae | L,B | Dec, Fod | Oral | IP, RD | 0.033 |
| 37 | *Boerhavia diffusa* L. | Nyctaginaceae | Wp | Fod | Oral | IP, EP, Ap, Sym | 0.031 |
| 38 | *Manihot esculenta* Crantz | Euphorbiaceae | L | Fod | Oral | IP, RD, Ap | 0.031 |
| 39 | *Prosopis africana* (Guill. & Perr.) Taub. | Leguminosae | B | Dec, Powder | Oral | IP, RD, Sym | 0.031 |
| 40 | *Ocimum basilicum* L. | Lamiaceae | L | Dec, Fod | Oral | DD | 0.028 |
| 41 | *Pseudocedrela kotschyi* (Schweinf.) Harms | Meliaceae | Wp | Dec, Fod | Oral | IP | 0.028 |
| 42 | *Monotes kerstingii* Gilg | Dipterocarpaceae | B | Powder | Oral | RD | 0.025 |
| 43 | *Pennisetum polystachion* (L.) Schult. | Poaceae | Wp | Fod | Oral | Ap | 0.025 |
| 44 | *Senna occidentalis* (L.) Link | Leguminosae | L | Fod | Oral | IP, BD | 0.025 |
| 45 | *Allium sativum* L. | Alliaceae | Bu | Inf | Nasal route | BD | 0.019 |
| 46 | *Bridelia ferruginea* Benth. | Euphorbiaceae | B | Dec | Oral | IP | 0.019 |
| 47 | *Mangifera indica* L. | Anacardiaceae | L,B | Dec, Fod | Oral | DD, EP, BD, RD | 0.019 |
| 48 | *Psidium guajava* L. | Myrtaceae | L | Fod, Powder | Oral | IP, DD | 0.019 |
| 49 | *Citrus aurantifolia* (Christm. & Panzer) Swingle | Rutaceae | L,Fr | Fod, Powder | Oral | IP, DD, EP, BD | 0.017 |
| 50 | *Crateva adansonii* DC. | Capparaceae | L, Ls | Fod | Oral | DD | 0.017 |
| 51 | *Nicotiana tabacum* L. | Solanaceae | L | Tri, Fod | Oral | EP | 0.017 |
| 52 | *Strychnos spinosa* Lam | Loganiaceae | L | Fod | Oral | IP | 0.017 |
| 53 | *Borassus aethiopum* Mart. | Palmae | L,Fr | Tri | Oral | DD, Ap | 0.014 |
| 54 | *Cajanus cajan* (L.) Millsp. | Leguminosae | L | Fod | Oral | IP, EP | 0.014 |
| 55 | *Calotropis procera* (Aiton) R.Br. | Asclepiadaceae | L | Fod | Oral | RD | 0.014 |
| 56 | *Chenopodium ambrosioides* L. | Chenopodiaceae | Ls | Fod | Oral | IP | 0.014 |
| 57 | *Combretum glutinosum* Perr. | Combretaceae | L,Fr | Fod, Powder | Oral | IP, Sym | 0.014 |
| 58 | *Luffa cylindrica* (L.) M.Roem. | Cucurbitaceae | L | Tri | Dermal route | IP, EP | 0.014 |
| 59 | *Panicum maximum* Jacq. | Poaceae | L | Fod | Oral | IP, DD, Sym | 0.014 |
| 60 | *Parkia biglobosa* (Jacq.) R. Br. ex G.Don | Mimosaceae | B | Dec, Powder | Oral | IP, BD | 0.014 |
| 61 | *Vitex doniana* Sweet | Verbenaceae | B | Dec | Oral | IP, EP | 0.014 |
| 62 | *Aganope stuhlmannii* (Taub.) Adema | Fabaceae | B | Dec | Oral | DD | 0.011 |
| 63 | *Anogeissus leiocarpa* (DC.) Guill. & Perr. | Combretaceae | B | Dec, Powder | Oral | IP | 0.011 |
| 64 | *Ficus umbellata* Vahl | Moraceae | L | Fod | Oral | DD | 0.011 |
| 65 | Heliotropium indicum L. | Boraginaceae | Wp | Fod | Oral | EP | 0.011 |
| 66 | *Indigofera spicata* Forssk | Leguminosae | L,B | Fod | Oral | IP, DD | 0.011 |
| 67 | *Paullinia pinnata* L. | Sapindaceae | L,B | Dec, Fod | Oral | Sym | 0.011 |

**Table 2.** *Cont.*

| N° | Species | Families | Parts Used | Preparation Methods | Administration Route | Diseases Treated | UV |
|----|---------|----------|------------|---------------------|----------------------|------------------|-----|
| 68 | *Vossia cuspidata* (Roxb.) Griff. | Poaceae | L | Fod | Oral | DD | 0.011 |
| 69 | *Annona muricata* L. | Annonaceae | B | Dec | Oral | IP | 0.008 |
| 70 | *Citrus limon* (L.) Burm. f. | Rutaceae | L,Fr | Tri, Fod, Powder | Oral, Instillation | EP, K | 0.008 |
| 71 | *Cocos nucifera* L. | Arecaceae | B | Dec | Oral | Ap | 0.008 |
| 72 | *Cucurbita moschata* Duchesne | Cucurbitaceae | L | Fod | Oral | IP | 0.008 |
| 73 | *Ficus platyphylla* Delile | Moraceae | L | Fod | Oral | Sym | 0.008 |
| 74 | *Jatropha gossypiifolia* L. | Euphorbiaceae | L | Mac, Fod | Oral | IP | 0.008 |
| 75 | *Margaritaria discoidea* (Baill.) Webster | Euphorbiaceae | L,B | Dec, Tri, Fod | Oral, Instillation | K, Sym | 0.008 |
| 76 | *Securinega virosa* (Roxb. ex Willd.) Baill | Euphorbiaceae | L | Fod | Oral | IP | 0.008 |
| 77 | *Sida acuta* Burm.f. | Malvaceae | L | Fod | Oral | DD | 0.008 |
| 78 | *Tephrosia bracteolata* Guill. & Perr. | Leguminosae | L | Fod | Oral | IP, DD | 0.008 |
| 79 | *Anacardium occidentale* L. | Anacardiaceae | L,B | Dec | Oral | IP | 0.006 |
| 80 | *Ceiba pentandra* (L.) Gaertn. | Malvaceae | L | Fod | Oral | DD | 0.006 |
| 81 | *Citrullus lanatus* (Thunb.) Matsum. & Nakai | Cucurbitaceae | L | Fod | Oral | IP, DD | 0.006 |
| 82 | *Crossopteryx febrifuga* (G. Don) Benth. | Rubiaceae | L,B | Dec, Fod | Oral | RD | 0.006 |
| 83 | *Datura innoxia* Mill. | Solanaceae | L | Tri, Fod | Oral | IP | 0.006 |
| 84 | *Lophira lanceolata* Tiegh. ex Keay | Ochnaceae | L,B | Inf, Fod | Oral | IP | 0.006 |
| 85 | *Oryza sativa* L. | Poaceae | L | Dec | Oral | IP, DD | 0.006 |
| 86 | *Abelmoschus esculentus* (L.) Moench | Malvaceae | L | Fod | Oral | IP | 0.003 |
| 87 | *Aframomum melegueta* K. Schum. | Zingiberaceae | L | Tri | Oral | IP | 0.003 |
| 88 | *Caesalpinia pulcherrima* (L.) Sw. | Leguminosae | L | Fod | Oral | IP | 0.003 |
| 89 | *Citrus sinensis* (L.) Osbeck | Rutaceae | L,Fr | Fod, Powder | Oral | IP | 0.003 |
| 90 | *Euphorbia balsamifera* Aiton | Euphorbiaceae | L | Tri, Fod | Oral | DD | 0.003 |
| 91 | *Ficus sycomorus* L. | Moraceae | L | Fod | Oral | IP | 0.003 |
| 92 | *Flueggea virosa* (Roxb. ex Willd.) Voigt | Euphorbiaceae | L | Fod | Oral | DD | 0.003 |
| 93 | *Ipomoea batatas* (L.) Lam. | Convolvulaceae | L | Fod | Oral | IP | 0.003 |
| 94 | *Isoberlinia doka* Craib & Stapf | Caesalpiniaceae | B | Dec | Oral | BD | 0.003 |
| 95 | *Manihot glaziovii* Müll.Arg. | Euphorbiaceae | L | Fod | Oral | Ap | 0.003 |
| 96 | *Musa* sp. | Musaceae | L | Fod | Oral | IP | 0.003 |
| 97 | *Tamarindus indica* L. | Leguminosae | L | Dec, Fod | Oral | BD | 0.003 |
| 98 | *Trianthema portulacastrum* L. | Aizoaceae | L | Fod | Oral | IP | 0.003 |
| 99 | *Vitellaria paradoxa* C.F. Gaertn. | Sapotaceae | B | Dec | Oral | IP | 0.003 |

**Legend:** Parts used: L: leaves; B: bark; Ls: leafy stem; Wp: whole plant; Fr: fruits; R: root; S: seed; Bu: bulb. Preparation methods: Mac: maceration; Inf: infusion; Dec: decoction; Fod: fodder; Roast: roasted form. Pathologies: IP: internal parasites; DD: digestive disorders; EP: external parasites; BD: breathing disorders; RD: reproductive disorders; Ap: appetence; K: keratitis; Sym: general symptoms.

**Table 3.** Plants parts used by breeders.

| Plants Parts Used | Number (*n* = 360) | Percentage (%) |
|---|---|---|
| Leaves | 216 | 59.99 |
| Bark | 78 | 21.60 |
| Leafy stem | 20 | 5.60 |
| Whole plant | 14 | 4 |
| Fruits | 14 | 4 |
| Root | 6 | 1.60 |
| Seed | 6 | 1.60 |
| Spike | 3 | 0.80 |
| Bulb | 3 | 0.80 |

**Table 4.** Use methods of traditional remedies.

| | Number (*n* = 360) | Percentage (%) |
|---|---|---|
| **Methods of preparation** | | |
| Fresh leaves | 177 | 49.01 |
| Decoction | 74 | 20.53 |
| Powder | 48 | 13.25 |
| Trituration | 26 | 7.28 |
| Maceration | 19 | 5.3 |
| Infusion | 14 | 3.97 |
| Roasted form | 2 | 0.66 |
| **Route of administration** | | |
| Oral | 339 | 94.18 |
| Instillation | 10 | 2.91 |
| Cutaneous | 7 | 1.94 |
| Nasal | 4 | 0.97 |

*3.4. Quantitative Analysis*

3.4.1. Use Value (UV)

The results show that 50.51% of the species inventoried are used for only one illness, while 22.22% are used for two illnesses. Half of this percentage (11.11%) is used to treat three diseases or symptoms. Less than 10% of the plants listed are involved in the treatment of four diseases or symptoms (9.09%) and only 5.05% and 2.02% plants, respectively are involved in the treatment of 5–6 diseases or symptoms. The use value was calculated (Table 2) to measure the relative degree of use of each of the plants cited by the breeders. This value ranged from 0.484 to 0.003 and seven species had a Use Value above 0.2 for 7.07% of the plants listed. Only 18.18% of the species inventoried had a Use Value greater than 0.1, while all the rest (81.82%) had UV ranges from 0.003 to 0.092. The highest value is found by *Vernonia amygdalina* (UV = 0.486), followed by *Khaya senegalensis* (UV = 0.436), *Launaea taraxacifolia* (UV = 0.431), *Napoleonaea vogelii* (UV = 0.378), *Momordica charantia* (UV = 0.358), *Moringa oleifera* (0.356) and *Zanthoxylum zanthoxyloides* (0.269). Fourteen species had the lowest Use Value (UV = 0.003). These are *Abelmoschus esculentus*, Aframomum melegueta, *Caesalpinia pulcherrima*, *Citrus sinensis*, *Euphorbia balsamifera*, *Ficus sycomorus*, *Flueggea virosa*, *Ipomoea batatas*, *Isoberlinia doka*, *Manihot glaziovii*, *Musa* sp., *Tamarindus indica*, *Trianthema portulacastrum* and *Vitellaria paradoxa*. The UV > 20% of each of these seven plants compared to the others indicated that they were of significant use in the treatment of small ruminant diseases in the study area.

3.4.2. FCR, FL and IAR

A total of 63 medicinal plants with anthelmintic potential representing 58 genera were recorded as plants used to treat internal parasitosis disease in small ruminants (Table 5). According to breeders, the recognition of this disease is based on the behaviour of sick animals and the symptoms observed. The main signs are, among others, anorexia, taste

perversion, diarrhoea, weight loss, dullness of the hair, growth retardation, especially in young animals, and a decrease in reproductivity for animals of childbearing age.

**Table 5.** Plants used by breeders to treat gastrointestinal parasitosis of small ruminants in the study area.

| Number | Species | Citation | RFC (%) | FL (%) | IAR |
|---|---|---|---|---|---|
| 1 | *Khaya senegalensis* (Desr.) A.Juss. | 142 | 39.44 | 90.45 | 0.97 |
| 2 | *Launaea taraxacifolia* (Willd.) Amin ex C. Jeffrey | 116 | 32.22 | 74.84 | 0.99 |
| 3 | *Napoleonaea vogelii* Hook. & Planch | 103 | 28.61 | 75.74 | 0.99 |
| 4 | *Momordica charantia* L. | 89 | 24.72 | 68.99 | 0.98 |
| 5 | *Vernonia amygdalina* Delile | 78 | 21.67 | 44.57 | 0.97 |
| 6 | *Moringa oleifera* Lam. | 64 | 17.78 | 50 | 0.97 |
| 7 | *Detarium microcarpum* Guill. & Perr. | 62 | 17.22 | 100 | 1 |
| 8 | *Zanthoxylum zanthoxyloides* (Lam.) Zepern. & Timler | 60 | 16.67 | 61.86 | 0.97 |
| 9 | *Carica papaya* L. | 51 | 14.17 | 98.08 | 0.98 |
| 10 | *Morinda lucida* Benth. | 28 | 7.78 | 39.44 | 0.94 |
| 11 | *Azadirachta indica* A. Juss. | 27 | 7.50 | 50.94 | 0.92 |
| 12 | *Newbouldia laevis* Seem. ex Bureau | 27 | 7.50 | 69.23 | 0.92 |
| 13 | *Pterocarpus erinaceus* Poir. | 25 | 6.94 | 55.56 | 0.95 |
| 14 | *Caesalpinia bonduc* (L.) Roxb. | 21 | 5.83 | 91.30 | 0.91 |
| 15 | *Ocimum gratissimum* L. | 21 | 5.83 | 35.59 | 0.95 |
| 16 | *Piliostigma reticulatum* (DC.) Hochst. | 20 | 5.56 | 100 | 1 |
| 17 | *Sterculia setigera* Delile | 17 | 4.72 | 100 | 1 |
| 18 | *Hyptis suaveolens* (L.) Poit. | 12 | 3.33 | 60 | 0 |
| 19 | *Sarcocephalus latifolius* (Lam.) de Wit | 11 | 3.06 | 84.62 | 0.92 |
| 20 | *Annona senegalensis* Pers. | 10 | 2.78 | 33.33 | −0.50 |
| 21 | *Pseudocedrela kotschyi* (Schweinf.) Harms | 10 | 2.78 | 100 | 1 |
| 22 | *Mitragyna inermis* (Willd.) Kuntze | 9 | 2.50 | 75 | 0.91 |
| 23 | *Bridelia ferruginea* Benth. | 7 | 1.94 | 100 | 1 |
| 24 | *Strychnos spinosa* Lam | 6 | 1.67 | 100 | 1 |
| 25 | *Boerhavia diffusa* L. | 5 | 1.39 | 45.45 | 0.70 |
| 26 | *Chenopodium ambrosioides* L. | 5 | 1.39 | 100 | 1 |
| 27 | *Elaeis guineensis* Jacq. | 5 | 1.39 | 12.50 | 0.33 |
| 28 | *Anogeissus leiocarpa* (DC.) Guill. & Perr. | 4 | 1.11 | 100 | 1 |
| 29 | *Cajanus cajan* (L.) Millsp. | 4 | 1.11 | 80 | 0.75 |
| 30 | *Indigofera spicata* Forssk | 4 | 1.11 | 100 | 1 |
| 31 | *Senna occidentalis* (L.) Link | 4 | 1.11 | 44.44 | 0.88 |
| 32 | *Zea mays* L. | 4 | 1.11 | 28.57 | 0.92 |
| 33 | *Annona muricata* L. | 3 | 0.83 | 100 | 1 |
| 34 | *Combretum glutinosum* Perr. | 3 | 0.83 | 60 | 0.75 |
| 35 | *Cucurbita moschata* Duchesne | 3 | 0.83 | 100 | 1 |
| 36 | *Jatropha gossypiifolia* L. | 3 | 0.83 | 100 | 1 |
| 37 | *Manihot esculenta* Crantz | 3 | 0.83 | 27.27 | 0.80 |
| 38 | *Parkia biglobosa* (Jacq.) R. Br. ex G.Don | 3 | 0.83 | 60 | 0.75 |
| 39 | *Piliostigma thonningii* (Schumach.) Milne-Redh. | 3 | 0.83 | 18.75 | 0.80 |
| 40 | *Psidium guajava* L. | 3 | 0.83 | 42.86 | 0.83 |
| 41 | *Securinega virosa* (Roxb. ex Willd.) Baill | 3 | 0.83 | 100 | 1 |
| 42 | *Vitex doniana* Sweet | 3 | 0.83 | 60 | 0.75 |
| 43 | *Anacardium occidentale* L. | 2 | 0.56 | 100 | 1 |
| 44 | *Chromolaena odorata* (L.) R. King & H. Rob. | 2 | 0.56 | 12.50 | 0.67 |
| 45 | *Datura innoxia* Mill. | 2 | 0.56 | 100 | 1 |
| 46 | *Lophira lanceolata* Tiegh. ex Keay | 2 | 0.56 | 100 | 1 |
| 47 | *Luffa cylindrica* (L.) M.Roem. | 2 | 0.56 | 40 | 0.75 |
| 48 | *Prosopis africana* (Guill. & Perr.) Taub. | 2 | 0.56 | 18.18 | 0.80 |
| 49 | *Senna alata* (L.) Roxb. | 2 | 0.56 | 11.76 | 0.88 |
| 50 | *Tephrosia bracteolata* Guill. & Perr. | 2 | 0.56 | 66.67 | 0.50 |
| 51 | *Abelmoschus esculentus* (L.) Moench | 1 | 0.28 | 100 | - |
| 52 | *Aframomum melegueta* K. Schum. | 1 | 0.28 | 100 | - |
| 53 | *Caesalpinia pulcherrima* (L.) Sw. | 1 | 0.28 | 100 | - |
| 54 | *Citrullus lanatus* (Thunb.) Matsum. & Nakai | 1 | 0.28 | 50 | - |

**Table 5.** *Cont.*

| Number | Species | Citation | RFC (%) | FL (%) | IAR |
|---|---|---|---|---|---|
| 55 | *Citrus aurantifolia* (Christm. & Panzer) Swingle | 1 | 0.28 | 16.67 | - |
| 56 | *Citrus sinensis* (L.) Osbeck | 1 | 0.28 | 100 | - |
| 57 | *Ficus sycomorus* L. | 1 | 0.28 | 100 | - |
| 58 | *Ipomoea batatas* (L.) Lam. | 1 | 0.28 | 100 | - |
| 59 | *Musa* sp. | 1 | 0.28 | 100 | - |
| 60 | *Oryza sativa* L. | 1 | 0.28 | 50 | - |
| 61 | *Panicum maximum* Jacq. | 1 | 0.28 | 20 | - |
| 62 | *Trianthema portulacastrum* L. | 1 | 0.28 | 100 | - |
| 63 | *Vitellaria paradoxa* C.F. Gaertn. | 1 | 0.28 | 100 | - |

FCR: Relative Citation Frequency; FL: Fidelity Level; IAR: Informant Agreement Rate.

Considering Table 5, five species stood out with citation frequencies above 20%. These were *Khaya senegalensis* (39.44%), *Launaea taraxacifolia* (32.22%), *Napoleonaea vogelii* (28.61%), *Momordica charantia* (24.72%) and *Vernonia amygdalina* (21.67%). Of the first five species cited, *Khaya senegalensis* showed a fidelity level of about 90.45% followed by *Launaea taraxacifolia* (74.84%) and *Napoleonaea vogelii* (75.74%). *Momordica charantia* and *Vernonia amygdalina* had fidelity levels of 68.99% and 44.57%, respectively. *Launaea taraxacifolia* and *Napoleonaea vogelii* were among the five species mentioned above with significant therapeutic indices, with each having an Informant Agreement Rate of 99%. They were well known to the different breeders.

### 3.4.3. Informant Consensus Factor (IFC) on Diseases Affecting Livestock

In total, five main diseases (Table 6) were known to the livestock breeders in the study area that weakened the zootechnical performance of the small ruminants in their livestock. Several diseases were affecting the zootechnical performance of small ruminants in the study area. The calculation of the Informants' Consensus Factor (ICF) showed that breeders had a strong consensus on the diseases that caused losses in their livestock. The high values of the ICF, gastrointestinal parasitism (ICF = 0.94) and digestive disorders (ICF = 0.91) showed that the same conditions were cited by several breeders regardless of the area surveyed, indicating that small ruminant health problems were common to breeders in the study area. Intestinal worms are a real health problem for small ruminants. They were a major constraint to the development of the small ruminant livestock sector and their efficient and sustainable management is still a serious problem.

**Table 6.** Main diseases affecting breeders' livestock.

| Pathologies | ICF |
|---|---|
| Gastrointestinal parasitism | 0.94 |
| Digestive disorders | 0.91 |
| External parasitism | 0.82 |
| Breathing disorders | 0.81 |
| Reproductive disorders | 0.89 |

### 3.4.4. Correlation between Use Values and Relative Citation Frequency

Pearson's correlation was calculated to determine the relationship between use values and Relative Citation Frequency (RFC). The result shows that the variables UV and RFC are significantly correlated (Pearson's r = 0.94548) with a coefficient of determination $r^2 = 0.89392$ (Figure 3).

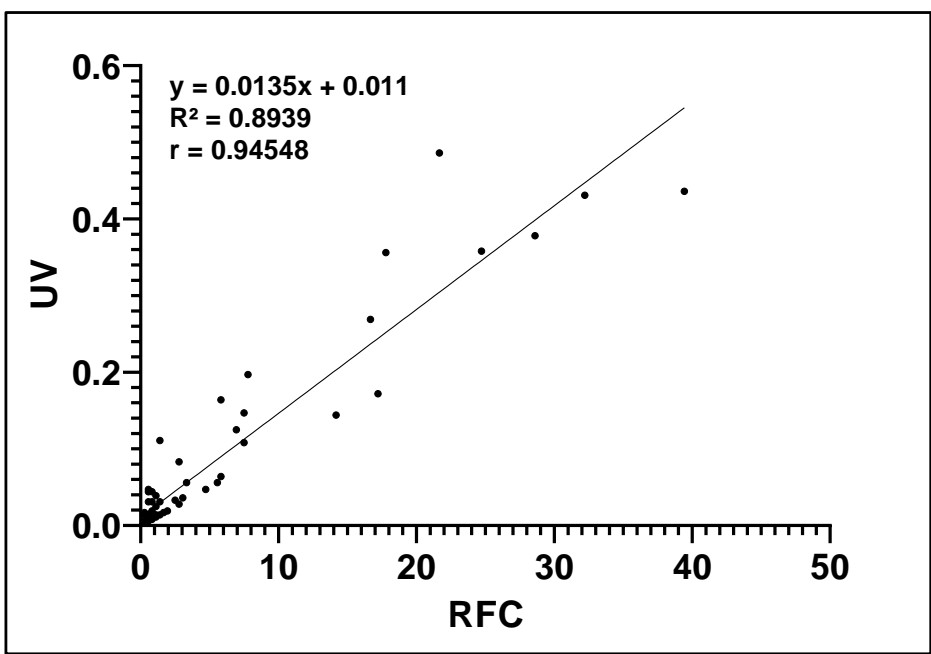

**Figure 3.** Correlation between UV and RFC.

### 3.4.5. Jaccard Index and Floristic Richness

The values of the Jaccard index ranged from 6.75% to 20.16%, and the highest value was observed in a study conducted in the cotton zone in North Benin, while the lowest degree was observed in the southern part of the same country. This result shows that the different studies are clearly independent (JI < 50%). Floristic richness was determined by considering the total flora of Benin, estimated at 2807 species according to Akoègninou et al. [12], and varies from 0.78 to 2.67, with the highest value obtained in the work of Hounzangbé-Adoté [6] in Southern Benin (Table 7).

**Table 7.** Comparison between the present study and some previous similar studies in Benin.

| Previous Study Area | (a) | (b) | (c) | JI (%) | Total Flora Estimated | Floristic Richness (%) | References |
|---|---|---|---|---|---|---|---|
| Southern Benin | 75 | 99 | 11 | 6.75 | 2807 | 2.67 | Hounzangbé-Adoté, [6] |
| Southern Benin | 22 | 99 | 15 | 14.15 | 2807 | 0.78 | Attindéhou et al. [7] |
| Cotton zone in North Benin | 56 | 99 | 26 | 20.16 | 2807 | 2 | Dassou et al. [8] |

(a): Species exclusive in another studies; (b): total species in the present study; (c): plants common to both areas; (JI): Jaccard index.

### 3.5. Correlation between Disease Variables and Factorial Axes

Table 8 shows four dimensions with varying proportions of the correlation matrix of the Principal Component Analysis (PCA). The cumulative proportions of the dimensions are 52.164%, 79.486%, 98.322% and 100% for dimensions 1, 2, 3 and 4, respectively. This translates that the information on internal and external parasites, digestive disorders and reproductive disorders is contained in all four dimensions. Interpretation of the eigenvalues of the aforesaid matrix shows that the first three dimensions explained 98.32% of the disease variability. As this information sharing is well above 50%, these first three dimensions can be used to adequately interpret the results of the PCA.

**Table 8.** Summary of models of the first 4 factorial axes.

| Dimensions | 1 | 2 | 3 | 4 |
|---|---|---|---|---|
| Eigenvalues | 2.087 | 1.093 | 0.753 | 0.067 |
| % of Variance | 52.164 | 27.322 | 18.837 | 1.678 |
| Cumulative % of Variance | 52.164 | 79.486 | 98.322 | 100.0 |

The study of the correlation between the three dimensions and the initial variables (Table 9) indicated that the variables "Internal parasites" and "Digestive disorders" were positively correlated with axis 1, which explains 52.16% of the variability (Figure 4). Thus, axis 1 reflects an involvement of the same plants practically in the treatment of internal parasitosis and digestive disorders. Axis 2, which explained 27.32% of the variability in pathologies (Figure 5), showed a negative correlation with the variables "External parasites" and "Reproductive disorders". Axis 2 showed that different plants were used to treat external parasites and reproductive disorders.

**Table 9.** Correlation between variables and factorial axes.

|  | Internal Parasites | Digestive Disorders | External Parasites | Reproductive Disorders |
|---|---|---|---|---|
| Dim.1 | 0.958 | 0.957 | 0.096 | 0.493 |
| Contribution | 43.993 | 43.910 | 0.441 | 11.657 |
| $Cosinus^2$ | 0.918 | 0.916 | 0.009 | 0.243 |
| Dim.2 | 0.123 | 0.067 | 0.883 | −0.542 |
| Contribution | 1.394 | 0.412 | 71.340 | 26.854 |
| $Cosinus^2$ | 0.015 | 0.005 | 0.780 | 0.293 |
| Dim.3 | −0.182 | −0.215 | 0.459 | 0.681 |
| Contribution | 4.397 | 6.109 | 28.005 | 61.489 |
| $Cosinus^2$ | 0.033 | 0.046 | 0.211 | 0.463 |

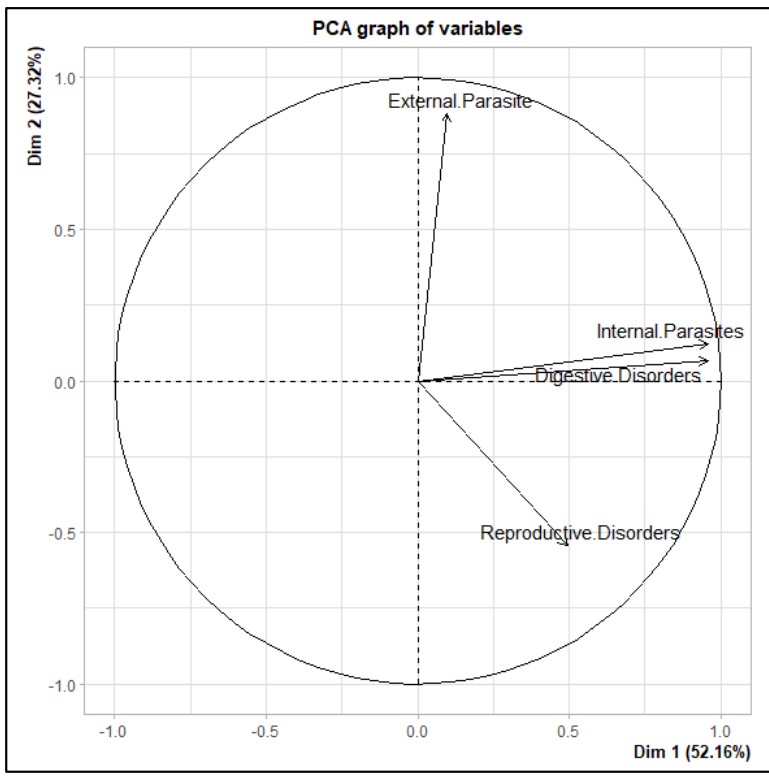

**Figure 4.** Correlation circle of PCA variables.

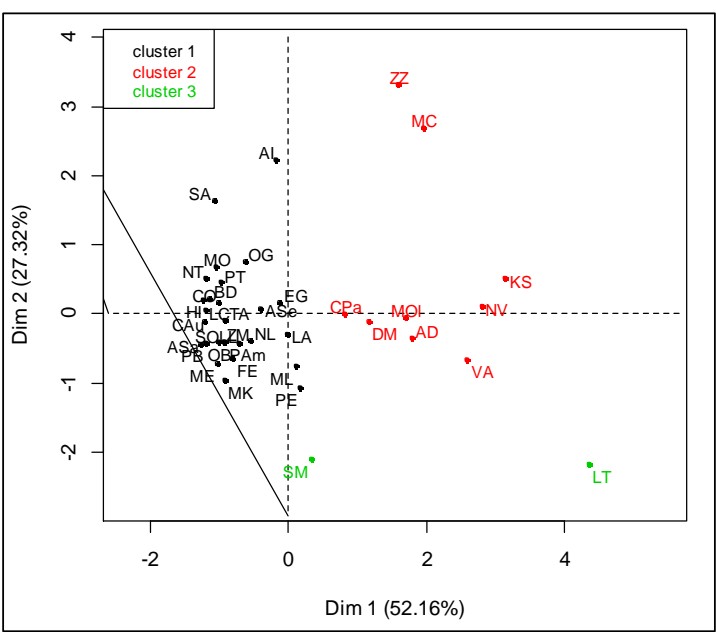

**Figure 5.** Positioning of species in the axis 1 and 2 system.

Table 9 presents the correlation between the variables and the factorial axes. From this table, a highly positive correlation between internal parasitism (0.958 > 0.05) and digestive disorders (0.957 > 0.05) was observed for dimension 1. Thus, on the farms, internal parasitism problems were followed by digestive disorders. This high correlation also indicated that the same plants were used by the breeders/agro-pastoralists to solve the problems of these two diseases. In dimension 2, the variables external parasitism and reproductive disorder were negatively correlated. Thus, plants that were used to control external parasites were not used to treat reproductive disorders. On the other hand, in dimension 3, a positive correlation was observed with the variables internal parasitism and reproductive disorders. Both dimensions 2 and 3 showed a negative correlation between internal parasitism (0.883) and reproductive disorders (−0.542) and a positive correlation for the same pathologies, respectively.

### 3.6. Hierarchical Classification of Anthelmintic Plants According to Groups

The projection of the different observations into the axis 1 and 2 system indicated that the members of Group 1 (71.8% of the plants) were mainly located in the negative part of axis 1 (Figure 5). The plants *Chromolaena odorata* (CO), *Luffa cylindrica* (LC), *Terminalia avicennioides* (TA), *Boerhavia diffusa* (BD), *Citrus aurantifolia* (CAu) *Senna alata* (SA), *Allium sativum* (Asa), *Monotes kerstingii* (MK) *Senna occidentalis* (SO) and *Parkia biglobosa* (PB) were the most commonly used plants in Group 1, whereas *Adansonia digitata* (AD), *Detarium microcarpum* (DM), *Moringa oleifera* (MOl), *Carica papaya* (CPa), *Vernonia amygdalina* (VA), *Khaya senegalensis* (KS), *Zanthoxylum zanthoxyloides* (ZZ), *Napoleonaea vogelii* (NV), *Momordica charantia* (MC) and *Adansonia digitata* (AD) were more commonly used in Group 2. Only the plant species *Launaea taraxacifolia* (LT) and *Spondias mombin* (SM) were used by Group 3. The 28 species constituting Group 1 largely dominate Groups 2 and 3 in the treatment of internal parasitosis and digestive disorders of small ruminants (Figure 6).

The analysis of the results of the PCA performed on pathologies and medicinal plants revealed that the first principal component alone accounted for 52.16% (>50%) of the input information, which was sufficient to ensure accuracy of interpretation. However, the first two principal components, expressing 90.91% of the input information, were retained for the analysis of results. The variables are grouped into classes in Table 10 and the following three classes emerged: class 1 (71.8%) class 2 (23.1%) and class 3 (5.1%). From this table, it appeared that the variables "internal parasites" and digestive disorders were predominant and characterised class 1. Class 2 was dominated by internal parasitism at

72.11%, followed by digestive disorders (21.11%). Class 3 was much more characterised by internal parasitism. Class 2 was characterised by digestive disorders and class 3 by reproductive disorders.

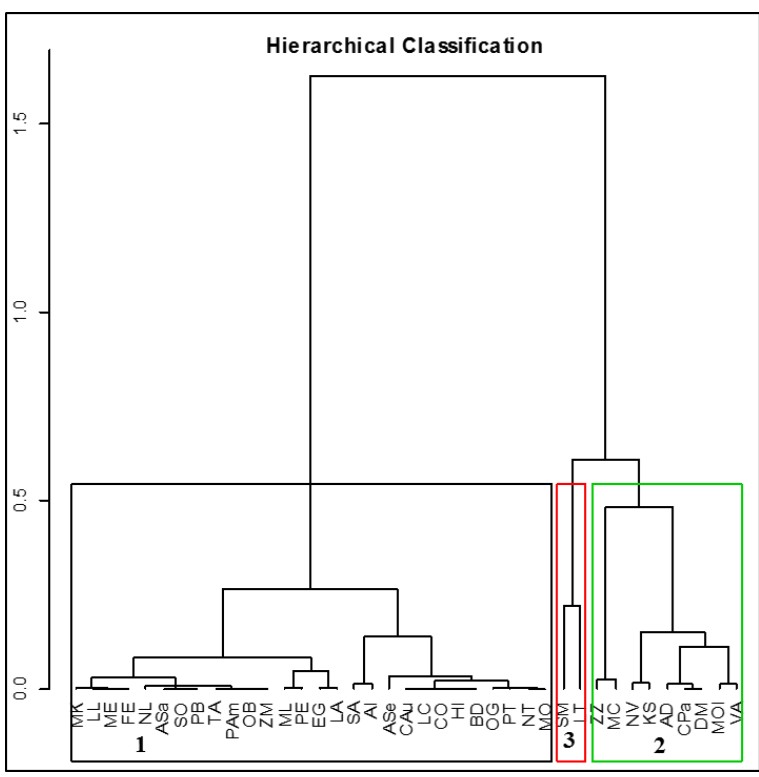

**Figure 6.** Hierarchical ascending classification of anthelmintic plants according to groups.

**Table 10.** Classification table of variables.

| Class | Class 1 (71.8%) | Class 2 (23.1%) | Class 3 (5.1%) | Total | Statistic F | *p* Value | Significance |
|---|---|---|---|---|---|---|---|
| Internal parasites | 6.14 a ± 9.571 | 72.11 b ± 38.921 | 58 b ± 82.024 | 24.03 ± 37.484 | 27.08 | 0.000 | *** |
| Digestive disorders | 6.54 a ± 9.968 | 21.11 b ± 21.468 | 5 ab ± 7.071 | 9.82 ± 14.433 | 4.194 | 0.023 | * |
| External parasites | 3.11 ± 4.149 | 5.78 ± 8.423 | 1 ± 1.414 | 3.62 ± 5.373 | 1.096 | 0.345 | ns |
| Reproductive disorders | 2.21 a ± 3.814 | 4.78 a ± 7.259 | 36.5 b ± 3.536 | 4.56 ± 8.917 | 47.78 | $7.39 \times 10^{-11}$ | *** |

Letters a, b on the same line indicate significant differences between means at the 5% level. ns = not significant ($p > 0.05$); * $p < 0.05$; *** $p < 0.001$.

## 4. Discussion

### 4.1. Methodology Used and Choice of Study Area

This study was conducted in the 12 communes of the cotton zone of Central Benin, and provided a database on the ethno-veterinary approach to medicinal plants used by the local population to treat small ruminants. The study was carried out with 360 herders in the form of individual interviews with a survey form that had been pre-tested during an exploratory survey. The choice of the study area was motivated by the absence of ethnoveterinary studies related to medicinal plants used to treat gastrointestinal parasitosis diseases in small ruminants. Some ethnomedical veterinary studies have already been carried out in Benin. Examples include the work carried out by Hounzangbé-Adoté [6] and Attindéhou [7] in southern Benin on medicinal plants used to treat pathologies in small ruminants. In the north of Benin, we can cite the work carried out by Dassou et al. [8]. However, no study has been carried out specifically on medicinal plants used in the treatment of small ruminant gastrointestinal parasitosis disease in all the communes of the study area, even though this zone is the largest and is considered to be dominated by small ruminant breeding practices compared to other area [2,10].

## 4.2. Characteristics of Respondents

In this ethno-veterinary study, of the 360 people interviewed, the vast majority were male. This profile is typical of most ethno-veterinary studies. This is due to the fact that animal herds are usually kept by men. Similar results are observed in the country by Dassou et al. [8] and Ouachinou et al. [9]. On the other hand, in an ethnomedicine study conducted by Kefifa et al. [23] in the semi-arid region of Algeria, the gender distribution of respondents does not match that of the present study. A female predominance was found, with 69% of women using medicinal plants to treat ailments. This finding could be linked to the type of ethnomedicine study concerned, as in some cultures and traditions, household well-being and health care are the prerogative of women.

The majority of the respondents were aged between 40 and 60 years. Older age is therefore an assurance of knowledge and mastery of ethnoveterinary practices. Our results are consistent with the work of Ouachinou et al. [24], who found that respondents aged 40–60 years were in the majority in the study population. Other studies conducted by Kefifa et al. [23] and Houndje et al. [25], respectively, showed that the majority of respondents were in the 41–50 and 50–59 age groups. This suggests that older people constitute ethno-medical libraries, where knowledge of veterinary herbal medicine is retained for years.

## 4.3. Medicinal Plant Diversity, Part Used and Administration Methods

This investigation in the study area shows how important medicinal plants are to rural livestock keepers in the management of diseases in general, and gastrointestinal parasitosis of small ruminants in particular. The survey revealed a multitude of plants with therapeutic value. The floristic wealth recorded in the study area represents about 4% of the total Beninese flora. This value is double (about 2%) that obtained by Dassou et al. [8]. On the other hand, it is low compared to that obtained by Ouachinou et al. [9], but the majority of species recorded in that study are also found in the present work. Hounzangbé-Adoté [6] and Attindéhou et al. [7] identified fewer medicinal plants than those obtained in the present study. This difference in species richness may be due to the specific diversity varying from one phytogeographic zone or district to another within the country [26]. More than half (63.64%) of the plants listed are used to treat gastrointestinal parasites. Other authors, such as Koné and Kamanzi [27], have also gone in the same direction by listing plants for veterinary use in general and those specific to the treatment of intestinal parasites. They found similar results in the north of the Ivory Coast.

In this study, the most represented plant families were the Leguminosae. This family is well represented as it contains species such as *Pterocarpus erinaceus*, *Caesalpinia bonduc*, *Piliostigma reticulatum*, *Senna alata* and *Prosopis Africana*, which are widely used in veterinary pharmacopoeia. The studies of Dassou et al. [8] and Ouachinou et al. [9] corroborate this result but with varying percentages of Leguminosae. The present result is contrary to that obtained by Sema et al. [28] in Togo, and Kefifa et al. [23] in Algeria, who obtained, respectively, Fabaceae and Asteraceae as dominant botanical families. This difference could be attributed to the diversity of socio-cultural groups that vary from country to country, as the use of natural plant resources for therapeutic purposes can be correlated with the knowledge of socio-cultural groups. The ethnoveterinary knowledge of breeders in the cotton zone of Central Benin is diverse and rich because of the diversity of socio-cultural groups and plant species available. For the species identified in this study, all plant parts can be used and they are chosen according to the ailment. Leaves are widely used directly as fodder to treat animals. Other organs such as barks and roots are also used and are often prepared as a decoction or macerate. Several studies corroborate that in pharmacopoeia, it is mainly the leaves and bark that are used to prepare traditional remedies [9,28]. Leaves are more commonly used because the removal of this part of the plant does not affect the viability of the species [29] and is safe for the plant itself, unlike bark and roots. It is also the part of the plant that small ruminants like best. However, the indiscriminate removal of the useful organ can impact the chlorophyll assimilation process in plants and consequently

lead to the disappearance of some herbaceous species involved in traditional treatments, as highlighted by Bi et al. [30].

### 4.4. Analysis of the Quantitative Study

Ethnobotanical indices were calculated to assess the degree of use of the species recorded. The more uses a species has, the higher its Use Value. It is noted in the work of Barkatullah et al. [31] that the species with the highest value also has the highest frequency of citation in the study. This is not the case in the present study. *Vernonia amygdalina* is the plant with the highest Use Value (0.486) but is the fifth most cited plant by breeders. The FL rate below 50% could justify this fifth position when quoted by breeders. Indeed, *Vernonia amygdalina* is a widely used dietary supplement for human culinary purposes in Benin and in Africa in general, and its involvement in veterinary care may give rise to a conflict of interest if it is less available in the area. However, its high Use Value is justified by its involvement in the treatment of six different diseases. The Use Values and relative frequency of quotation, the level of loyalty and the degree of consensus of the informants showed that *Khaya senegalensis*, *Launaea taraxacifolia*, *Napoleonaea vogelii*, *Momordica charantia* and *Vernonia amygdalina* are very important in the management of small ruminant gastrointestinal parasitosis in the study area because they are among the most used plants and the most quoted by the breeders, who are consensual as to their usefulness and effectiveness. The Pearson correlation result showed a linear relationship between the Use Value (UV) and the Relative Frequency of Citation (RFC). In the present study, the UV margin is significantly lower than that obtained in the work conducted by Kefifa et al. [23], Barkatullah et al. [31] and Bano et al. [32]. These authors obtained, respectively, UV margins of the order of 0.03–1.24, 0.05–1.21 and 0.12–1.64. This clear difference could be explained by the circulation of therapeutic information according to the socio-cultural groups, the use made of the plants, the level of knowledge of the plants and their therapeutic indications, the prevalence of the diseases for which they are used, the accessibility and/or the availability of the species, the period of blooming and the socio-cultural prohibitions. This can be corroborated by the example of *Napoleonaea vogelii* Hook & Planch and *Vitellaria paradoxa* C.F. Gaertn., both used to treat internal parasites in small ruminants but with respective Use Values of 0.378 and 0.003. Of the two plants, one is better known to breeders than the other, despite the fact that they are both involved in the treatment of the same disease. The ICF calculation shows that values are above 50%. This indicates that informants have a higher consensus for the gastrointestinal parasites affecting their herds. A high value (close to 1) indicates that the same disease is well known to breeders throughout the study area. On the other hand, a low value indicates that informants do not agree on the pathologies affecting the herds. According to the literature, ICF values between 0.25 and 0.73 are significant [20]. The consensus factor values obtained after this study show the importance of controlling the gastrointestinal parasitosis diseases prevalent in the study environment. Other similar studies have obtained more or less different values. The values obtained by Ong and Kim [33] ranged from 0.75 to 1 and those of Faruque et al. [34] were 0.50–0.66. The latter values are much lower than those obtained in the present study. These differences could be explained by the prevalence of diseases affecting the animals in each study area.

The Pearson correlation coefficient r calculated in this study is r = 0.94 close to 1 with a coefficient of determination of $r^2 = 0.89$. The positive Pearson's r value indicates a significant correlation between UV and FRC [32]. The value of this coefficient is higher than that obtained by Barkatullah et al. [31], which was r = 0.8682 with $r^2 = 0.75$.

The values of the Jaccard index ranged from 6.75% to 20.16%. This result is different from that obtained by Faruque et al. [34] in Bangladesh. These authors had obtained a low value of 1.65 and the highest was 33. It should be noted that these authors considered a large number of previous studies for the calculation of the JI and had also obtained a higher number of medicinal species (117 species) than in the present study (99 species).

## 5. Conclusions

This study is the first to be carried out in the cotton zone of Central Benin. It does not show any proven similarity with previous studies conducted in the territory but proves that this zone has a very diverse range of medicinal anthelmintic plants used to treat small ruminants. In vitro and in vivo tests need to be carried out on some of these plants to confirm or refute their anthelmintic properties. This will provide a scientific database for the search for new anthelmintic molecules and for the development of improved traditional medicines.

**Author Contributions:** Conceptualization, C.C.D. and P.A.O.; methodology, C.C.D. and P.A.O.; software, C.C.D.; validation, P.A.O. and A.B.A.; formal analysis, C.C.D. and B.S.B.K.; investigation, C.C.D., O.S. and K.B.A.; resources, P.A.O.; data curation, C.C.D.; writing—original draft preparation, C.C.D.; writing—review and editing, C.C.D. and P.A.O.; visualization, L.L.; supervision, P.A.O. and S.M.H.-A. All authors have read and agreed to the published version of the manuscript.

**Funding:** This research received no external funding.

**Institutional Review Board Statement:** The study was conducted in accordance with the guidelines of the Declaration of Helsinki, and approved by the Ethics Committee of the National University of Agriculture of Porto Novo for research and code of practice for housing, care and welfare of animals used in scientific procedures, N° 143- 2018/ President-CER/SA of 8 November 2018.

**Informed Consent Statement:** Informed consent was obtained from all subjects involved in the study.

**Data Availability Statement:** Data are available from the authors upon reasonable request.

**Acknowledgments:** The authors are grateful to the Graduate School of Agricultural and Water Sciences of the National University of Agriculture for their continuous support. We are also grateful to all the breeders who participated in this study.

**Conflicts of Interest:** The authors declare no conflict of interest.

## Abbreviations

Ap: appetence; B: bark; BD: breathing disorders; Bu: bulbe; DD: digestive disorders; Dec: decoction; Dim: dimension; EP: external parasites; FCR: Relative Citation Frequency; FL: Fidelity Level; Fod: fodder; Fr: fruits; IAR: Informant Agreement Rate; JI: Jaccard index; Inf: infusion; IP: internal parasites; K: keratitis; L: leaves; Ls: leafy stem; Mac: maceration; R: root; RD: reproductive disorders; Roast: roasted form; S: seed; Sym: general symptoms; Wp: whole plant.

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
