# Peer review of "Ethno-Veterinary Survey and Quantitative Study of Medicinal Plants with Anthelmintic Potential Used by Sheep and Goat Breeders in the Cotton Zone of Central Benin (West Africa)"

_2571-8800, doi:10.3390/j4040040_

Round 1
Reviewer 1 Report
The article must be accepted.
The authors presented the findings of a survey "Ethno-Veterinary Survey and Quantitative Study of Medicinal Plants with Anthelmintic Potential Used by Sheep and Goat Breeders in Agro-Ecological Zone 5 of Benin (West Africa). These findings are worth publishing because they can considerably contribute to the knowledge about the potential of investigation of some plants.The topic fits into the scope of this journal. Because it could provide an information base for the search for new molecules with medicinal potential, the study provides definitely novel scientific information.
Author Response
Thank you for your positive feedback.
Reviewer 2 Report
The manuscript „Ethno-Veterinary Survey and Quantitative Study of Medicinal Plants with Anthelmintic Potential Used by Sheep and Goat Breeders in Agro-Ecological Zone 5 of Benin (West Africa)” described results of ethno-veterinary study conducted in Benin.
In my opinion, the article is quite chaotic and contains a lot of information that is not necessary for the whole story and is unrelated to the aim of the study. Furthermore, the aim of the study purpose of the study is not clearly formulated.
The demographic analysis of the small ruminants owners is conducted with too many details and the authors pay too much attention to them in the article. Therefore I wonder what is the correlation between e.g. religion and sex of the owners and anthelmintic parasitic drugs for ruminants.
Thus, I wonder if the article was submitted to the proper journal…
Conversely, the authors do not describe how the plants with anthelmintic potential were identified by the owners of ruminants. I found only very poor information that „ directly in the field with the help of the respondents”. Nowadays, we have many possibilities for species estimation so identification of the plants visual assessment is insufficient. Similarly, the preparation of plant remedies is also is too poorly described.
Furthermore, I suggest the authors the examination of animals to evaluate the effectiveness of the anthelmintic potential plant.
Author Response
Point 1: In my opinion, the article is quite chaotic and contains a lot of information that is not necessary for the whole story and is unrelated to the aim of the study. Furthermore, the aim of the study purpose of the study is not clearly formulated
Response 1: The article has been corrected and irrelevant elements have been removed. The objective has been well defined in the revision (lines 60-62).
Point 2: The demographic analysis of the small ruminants owners is conducted with too many details and the authors pay too much attention to them in the article. Therefore I wonder what is the correlation between e.g. religion and sex of the owners and anthelmintic parasitic drugs for ruminants.
Response 2:
Demographic analysis.
Socio-demographic characteristics have been reduced to the essentials. In the new version of our manuscript we have considered characteristics relating to sex, age and activities carried out by the herders (see Table 1, lines 220-222)
Point 3: Conversely, the authors do not describe how the plants with anthelmintic potential were identified by the owners of ruminants. I found only very poor information that „ directly in the field with the help of the respondents”. Nowadays, we have many possibilities for species estimation so identification of the plants visual assessment is insufficient. Similarly, the preparation of plant remedies is also is too poorly described.
Response 3: The method of identifying plants in the field has been improved. Lines 111-117. The medicinal plants mentioned were identified in the area where breeders usually collect them with note for each plant species collected. Specimens were collected and numbered on the spot, later identified using Benin’s analytical flora, according to Akoègninou et al. [12]. Additional identification was carried out by matching specimens with previously identified held in the National Herbarium of Benin. Photographic and video cameras were used for graphic documentation. The botanical names of the plant specimens were updated according to The World Flora Online.
Point 4: Furthermore, I suggest the authors the examination of animals to evaluate the effectiveness of the anthelmintic potential plant.
Response 4: This part is another chapter of our research work and is currently being evaluated.
Reviewer 3 Report
The manuscript describes an ethno-veterinary survey and a quantitative study of medicinal plants with anthelmintic potential used by sheep and goat breeders in an agro-ecological zone of Benin (West Africa).
The survey was carried out throughout face-to-face interviews of three hundred and sixty breeders. Furthermore, the floristic diversity of the surveyed area is presented. Overall, the manuscript is of interest, the methods and the statistic analysis suitable and the results sounding. However, I have several concerns that should be acknowledged before the manuscript should be suitable for publication.
The first concern is about the “agro-ecological Zone 5” in the title, in the summary, key words and in the introduction. For a reader not educated about the different zones of Benin, zone 5 has no meaning. The term of cotton zone of central Benin (zone 5) should be used instead of Zone 5.
In many parts of the manuscript there are statements referred to previous studies (e.g. line 194), but there is no suitable reference.
In the Discussion there is no need to repeat some statements already done in the results. As it is, the Discussion is too long and boring. It should be shortened avoiding as far as possible any repetition (see as an example Socio-demographic characteristics). Shortening this chapter will improve the readability of the chapter, being of advantage for the entire manuscript.
I can agree that the study can provide some information for new molecules with anthelmintic potential but the critical point is that no information on the output of the use of the medical plants is given. The sentence should be rephrased.
Finally, I suggest to add a list of abbreviations and its meaning to the end of the manuscript.
Minor concerns
IAR and FCR should be defined in the abstract and when cited for the first time.
Line 50: zone 5 not V
Line 70: was
Line 90: remove (03)
Line 225: 12 communes
Table 1: the title is repeated two times
Figure 2 should be removed as data is already described in the text and Table 1.
Table 2 needs a caption for the abbreviations
Table 5 needs references for previous studies
Figures 4, 5 and 6 should be substitute by 2 tables, much more easy to be read
Table 3: specify “number”
Table 5 needs some reference of the previous studies carried out in Benin
Table 6 needs a more clear explanation
Lines 360-363: specify the diseases
Line 523: 12
Table 8: reproduction troubles
Author Response
Point 1: The first concern is about the “agro-ecological Zone 5” in the title, in the summary, key words and in the introduction. For a reader not educated about the different zones of Benin, zone 5 has no meaning. The term of cotton zone of central Benin (zone 5) should be used instead of Zone 5.
Response 1 : The term Agro-ecological Zone 5 has been replaced by " cotton zone of Central Benin " throughout the document.
Point 2 : In many parts of the manuscript there are statements referred to previous studies (e.g. line 194), but there is no suitable reference.
Response 2 : Line 194 talks about CFI and a reference is attributed to this, reference [20] which is Heinrich, M.; Ankli, A.; Frei, B.; Weimann, C.; Sticher, O. Medicinal plants in Mexico: Healers' consensus and cultural importance. Soc. Sci. Med. 1998, 47, 1859-1871.
Point 3 : In the Discussion there is no need to repeat some statements already done in the results. As it is, the Discussion is too long and boring. It should be shortened avoiding as far as possible any repetition (see as an example Socio-demographic characteristics). Shortening this chapter will improve the readability of the chapter, being of advantage for the entire manuscript.
Response 3 : The Socio-demographic characteristics part of the discussion has been reduced (lines 494-510). In the discussion set, repetitions of results have been reduced as much as possible.
Point 4 : I can agree that the study can provide some information for new molecules with anthelmintic potential but the critical point is that no information on the output of the use of the medical plants is given. The sentence should be rephrased.
Response 4 : The sentence has been improved (lignes 607-610). In vitro and in vivo tests need to be carried out on some of these plants to confirm or refute their anthelmintics properties. This will provide a scientific database for the search for new anthelmintics molecules and for the development of improved traditional medicines.
Point 5 : Finally, I suggest to add a list of abbreviations and its meaning to the end of the manuscript.
Response 5 : The list of abbreviations has been added at the end of the manuscript (lines 613-619).
Abbreviation : Ap: Appetence; B: Bark; BD: Breathing Disorders; Bu: Bulbe; DD: Digestive Disorders; Dec: Decoction; Dim: Dimension; EP: External Parasites; FCR: Relative Citation Frequencies; FL: Fidelity Level; Fod: Fodder; Fr: Fruits; IAR: Informant Agreement Rate; JI : Jaccard Index ; Inf: Infusion; IP: Internal Parasites; K : Keratitis ; L: Leaves; Ls: Leafy stem; Mac: Maceration; R: Root; RD: Reproductive Disorders; Roast: Roasted form ; S: Seed; Sym: General symptoms; Wp: Whole plant.
Point 6 : Minor concerns
- IAR and FCR should be defined in the abstract and when cited for the first time.
IAR and FCR have been defined in the summary as required
- Line 50: zone 5 not V
Zone 5 has been removed.
- Line 70: was
This has been corrected (line 67)
- Line 90: remove (03)
(03) has been deleted (line 77)
- Line 225: 12 communes
12 added (line 209)
- Table 1: the title is repeated two times
The repetition of the "Table 1 heading" is deleted
- Figure 2 should be removed as data is already described in the text and Table 1.
Figure 2 is deleted
- Table 2 needs a caption for the abbreviations
List of abbreviations is added for Table 2 (lines 244-248).
- Table 5 needs references for previous studies
Table 5, now Table 7, contains references to the data mentioned (lines 388-390)
- Figures 4, 5 and 6 should be substitute by 2 tables, much more easy to be read
Figures 4, 5 and 6 are replaced by Tables 3 and 4 (lines 276-303)
- Table 3: specify “number”
"Number" was specified on Table 3 which became Table 5 (line 306)
- Table 5 needs some reference of the previous studies carried out in Benin
Data on studies conducted outside Benin have been removed
- Table 6 needs a more clear explanation
Table 6, now Table 8, has been better explained (lines 394-401).
Table 8 shows four dimensions with varying proportions of the correlation matrix of the Principal Component Analysis (PCA). The cumulative proportions of the dimensions are 52.164%, 79.486%, 98.322% and 100% for dimensions 1, 2, 3 and 4 respectively. This translates that the information on internal and external parasites, digestive disorders and reproductive disorders is contained in all four dimensions. Interpretation of the eigenvalues of the aforesaid matrix shows that the first three dimensions explained 98.32% of the disease variability. As this information sharing is well above 50%, these first three dimensions can be used to adequately interpret the results of the PCA.
- Lines 360-363: specify the diseases
The disease (gastrointestinal parasitosis of small ruminants) was specified in the title of Table 5 (line 304).
- Table 8: reproduction troubles
Reproduction troubles corrigé, tableau 10 (ligne 467-468)
Reproduction troubles corrected (line 467-468)
Round 2
Reviewer 2 Report
The authors addressed the reviews in a convincing manner and increased the scientific value of the article. In my opinion, the revised manuscript is suitable for publication.
However, I have an additional question - how parasitic diseases were found in animals that were treated with plants? Can you please clarify this issue and include information about it in the manuscript?
Author Response
Response to Reviewer 2 Comments
Point 1: The authors addressed the reviews in a convincing manner and increased the scientific value of the article. In my opinion, the revised manuscript is suitable for publication. However, I have an additional question - how parasitic diseases were found in animals that were treated with plants? Can you please clarify this issue and include information about it in the manuscript?
Response 1: According to breeders, the recognition of digestive parasitic diseases is based on the behaviour of sick animals and the symptoms observed. For them, the main signs of digestive parasites in animals are, among others, anorexia, taste perversion, diarrhoea, weight loss, dullness of the hair, growth retardation especially in young animals and a decrease in reproductivity for animals of childbearing age (lines 335-339).